# Utilization of Mental Health Provision, Epistemic Stance and Comorbid Psychopathology of Individuals with Complex Post-Traumatic Stress Disorders (CPTSD)—Results from a Representative German Observational Study

**DOI:** 10.3390/jcm13102735

**Published:** 2024-05-07

**Authors:** David Riedl, Hanna Kampling, Tobias Nolte, Christina Kirchhoff, Johannes Kruse, Cedric Sachser, Jörg M. Fegert, Harald Gündel, Elmar Brähler, Vincent Grote, Michael J. Fischer, Astrid Lampe

**Affiliations:** 1Ludwig Boltzmann Institute for Rehabilitation Research, 1100 Vienna, Austria; vincent.grote@rehabilitation.lbg.ac.at (V.G.); michael.fischer@reha-kitz.at (M.J.F.);; 2University Hospital of Psychiatry II, Department of Psychiatry, Psychotherapy, Psychosomatics and Medical Psychology, Medical University of Innsbruck, 6020 Innsbruck, Austria; 3Department of Psychosomatic Medicine and Psychotherapy, Justus Liebig University Giessen, 35392 Giessen, Germany; 4Anna Freud National Centre for Children and Families, London N1 9JH, UK; 5Research Department for Clinical, Educational and Heath Psychology University College London, London WC1E 6BT, UK; 6Department for Psychosomatic Medicine and Psychotherapy, Medical Center of the Philipps University Marburg, 35037 Marburg, Germany; 7Department of Child and Adolescent Psychiatry/Psychotherapy, Ulm University, 89075 Ulm, Germany; 8German Center for Mental Health (DZPG), Partner Site Ulm, 89075 Ulm, Germany; 9Department of Psychosomatic Medicine and Psychotherapy, University Ulm Medical Center, 89081 Ulm, Germany; 10Department of Psychosomatic Medicine and Psychotherapy, University Medical Center of the Johannes Gutenberg-University, 55131 Mainz, Germany; 11Department of Medical Psychology and Medical Sociology, University Hospital Leipzig, 04103 Leipzig, Germany; 12VAMED Rehabilitation Center Kitzbuehel, 6370 Kitzbuehel, Austria; 13VAMED Rehabilitation Montafon, 6780 Schruns, Austria

**Keywords:** complex PTSD, epistemic trust, psychopathology, utilization of mental health provision, treatment, personality functioning

## Abstract

**Background**: Complex post-traumatic stress disorder (CPTSD) is a severely debilitating recently added symptom cluster in the International Classification of Diseases (ICD-11). So far, only limited information on mental health treatment-uptake and -satisfaction of individuals with CPTSD is available. The aim of this study is to investigate these aspects in a representative sample of the German general population. **Methods**: Participants completed the International Trauma Questionnaire (ITQ) to identify participants with CPTSD, as well as questionnaires on mental health treatment uptake and satisfaction, adverse childhood experiences, anxiety, depression, working ability, personality functioning, and epistemic trust. **Results**: Of the included *n* = 1918 participants, *n* = 29 (1.5%) fulfilled the criteria for CPTSD. Participants with CPTSD had received mental health treatment significantly more often than participants with PTSD or depression (65.5% vs. 58.8% vs. 31.6%; *p* = 0.031) but reported significantly less symptom improvement (52.9% vs. 78.0% vs. 80.0%; *p* = 0.008). Lower levels of epistemic trust were associated with higher CPTSD symptoms (*p* < 0.001). **Conclusions**: Our study shows that while the vast majority of individuals with CPTSD had received mental health treatment, subjective symptom improvement rates are not satisfactory. CPTSD was associated with a broad number of comorbidities and impairments in functioning. Lower levels of epistemic trust may partially explain worse treatment outcomes.

## 1. Introduction

In 1992, the concept of complex post-traumatic stress disorder (CPTSD) was introduced by Judith Herman to delineate the particular set of symptoms individuals who have endured prolonged, recurrent, or multiple traumatic stressors frequently experience [1]. The newest edition of the International Classification of Diseases (ICD-11) by the World Health Organization (WHO) now incorporates the diagnostic category of CPTSD [2], which involves exposure to a traumatic event—defined as an extremely threatening or horrific event or series of events—and encompass three Post-Traumatic Stress Disorder (PTSD) symptom clusters (intrusive recollections, avoidance behavior, and heightened arousal). Furthermore, three supplementary symptom clusters are used to describe pervasive and chronic disruptions in self-organization (DSO). These include (a) difficulties in regulating emotions (e.g., heightened emotional reactivity, anger outbursts, feeling emotionally numb, or dissociated), (b) a pervasive negative self-concept (e.g., feeling diminished, defeated, or worthless; pervasive feelings of shame, guilt), and (c) challenges in establishing and maintaining interpersonal relationships (e.g., feeling distant from others, having difficulty maintaining intimate relationships) [3]. A recent epidemiological study in the general population of Germany reported a one-month prevalence rate of 0.5% for CPTSD [4].

While the evidence base on CPTSD is still limited, research has indicated that CPTSD tends to be a more debilitating condition than PTSD [5]. Particularly in cases of early or prolonged abuse, affected individuals may develop severe disorders related to their self-perception, accompanied by feelings of helplessness, apathy, shame, guilt, and self-criticism, as well as a deep aversion to one’s own body, a feeling of being violated, and a strong self-disdain. This frequently results in significant self-neglect, hindered ability to form relationships, recurring relationship failures, withdrawal from social interactions, and enduring distrust toward others [6]. Affected individuals tend to show severe deficits in their mentalizing ability, which is defined as a mental process that facilitates the understanding and representation of intentional mental states in oneself and others by taking into account one’s own thoughts, needs, emotions, wishes, and desires as well as those of others [7,8,9]. It has been argued that epistemic isolation—which is associated with ineffective mentalizing alongside social trauma—may cause the individual to reject the content of new information, confuse its meaning, or even misinterpret it as being malignant [10]. This so-called state of epistemic hypervigilance creates inflexibility in adapting to changed situations and undermines effective social communication and learning [11]. Fonagy, Luyten [12] proposed that epistemic trust, i.e., the ability to assess the trustworthiness, relevance, and generalizability of information provided by others, aids in acquiring new knowledge, thus promoting social adaptability and resilience when faced with difficult information. However, individuals who have undergone childhood adversity and lack a secure attachment system may display increased levels of epistemic disruptions in the form of epistemic mistrust and epistemic credulity. Epistemic mistrust denotes a disposition to perceive all sources of information as untrustworthy or potentially malicious, leading to a reluctance to be swayed by external influences. Conversely, epistemic credulity manifests as a lack of discernment and skepticism regarding the reliability of information, rendering individuals more susceptible to misinformation and exploitation [13].

The vast majority of patients with CPTSD have a broad range of psychopathological comorbidities, including depressive disorders, anxiety disorders, or substance abuse disorders [14]. Consequently, individuals with CPTSD may need more advanced interventions that differ significantly in both quantity and quality from the treatments typically available for PTSD [15]. A review of psychotherapeutic interventions for CPTSD has shown that traditional interventions were associated with poorer treatment outcomes than other forms of trauma [16]. While there is a growing body of evidence regarding the epidemiology and clinical characteristics of individuals with CPTSD, information regarding psychotherapy uptake and treatment satisfaction is still scarce. It may be worthwhile to reiterate that our current understanding of the behavioral patterns exhibited by these patients remains limited, and population-based data could shed light on their propensity to seek treatment. This aspect is particularly crucial for this disorder, given the prevalent challenges in interpersonal interactions and trust deficits. Consequently, there arises a pertinent question regarding their inclination to undergo therapy, where both trust-building and interactive engagement are imperative. Additionally, there is evidence that CPTSD may still be underdiagnosed within the healthcare system [17], which makes access to appropriate treatment more difficult. In our study, we also assessed the patients’ epistemic trust and personality functioning in order to better characterize this group and thus identify approaches for treatment offers.

The aim of the present study, therefore, was to establish the overall expression of PTSD symptom clusters in a population-based sample and to present data on mental health treatment uptake and treatment satisfaction, as well as clinical characteristics and comorbidity of individuals with CPTSD. We specifically hypothesized that participants with CPTSD would (a) show significantly worse comorbid symptoms and personality functioning and (b) lower treatment satisfaction than participants with a history of mental health treatment who did not fulfill the criteria for CPTSD or participants without a history of mental health treatment.

## 2. Methods

### 2.1. Study Design and Participants

Representative data on the German population were collected by the demography research institute USUMA Berlin between December 2020 and March 2021. Using the reference system for representative studies in Germany provided by the ADM-Sampling-System, 258 German regional areas were predefined, and target households within these regional areas were selected following a random route procedure. For multi-person households, the Kish selection grid technique was used to randomly select one person. By these means, a total of N = 2519 randomly selected persons could be included, resulting in a representative sample of the German population. All participants completed face-to-face interviews to assess sociodemographic and personal characteristics as well as self-report questionnaires. Inclusion criteria comprised sufficient German language skills, being older than 16, and informed consent (for minors, informed consent was additionally obtained from a parent/legal guardian). The survey was conducted in accordance with the Declaration of Helsinki and fulfilled the ethical guidelines of the International Code of Marketing and Social Research Practice of the International Chamber of Commerce and the European Society of Opinion and Marketing Research. Adherence to all applicable hygiene regulations with regard to the COVID-19 pandemic was given. Ethical approval was obtained by the Ethics Committee of the Medical Faculty of the University of Leipzig (no. 474/20-ek, on 17 November 2020).

### 2.2. Measures

#### 2.2.1. Adverse Childhood Experiences Questionnaire (ACE)

The Adverse Childhood Experiences Questionnaire [18,19] is a commonly utilized self-assessment tool employed for the retrospective evaluation of various adversities experienced during early childhood [20]. Comprising ten items, where responses are coded as ‘yes’ (1) or ‘no’ (0), the ACE questionnaire encompasses areas such as emotional, physical, and sexual abuse, emotional and physical neglect, parental separation, witnessing violence against a parent, as well as familial challenges like substance use, mental illness, and incarceration, resulting in a cumulative score ranging from 0 to 10. Within our sample, we observed a strong internal consistency among the ACE items, with a coefficient alpha of 0.81.

#### 2.2.2. International Trauma Questionnaire (ITQ)

The ITQ [21] is a self-reported questionnaire designed to evaluate symptoms associated with post-traumatic stress disorder (PTSD) and CPTSD. The PTSD scale consists of six items, assessing the three main aspects of PTSD (re-experiencing, avoidance, and persistent perception of heightened current threat) along with a three-item scale to evaluate the functional impairment caused by these symptoms. For CPTSD assessment, an additional six items are included to evaluate the three core elements of disturbances in self-organization (DSO) (affective dysregulation, negative self-concept, and problematic relationships). The combination of PTSD and DSO symptoms allows the classification of CPTSD. Furthermore, three additional items for both CPTSD and DSO assess the functional impairment associated with each syndrome.

All items are rated on a five-point Likert scale from ‘not at all’ to ‘very much’. To maintain consistency with previous research, separate subscales for PTSD and DSO were calculated as recommended [22]. The PTSD and DSO symptom scores, therefore, range from 0 to 24, and the total ITQ scores range from 0 to 48. Symptoms were considered as present if a score of 2 or higher was reported by the participant. A diagnosis of PTSD was assigned if at least one of two symptoms of each of the three PTSD clusters was scored as present, along with at least one functional impairment cluster associated with PTSD was reported. A diagnosis of CPTSD was assigned if the diagnostic criteria for PTSD were met, and at least one symptom from each of the three DSO clusters was scored as present, along with at least one functional impairment cluster associated with CPTSD [21]. Consistent with the diagnostic rules of ICD-11, participants could receive a diagnosis of either PTSD or CPTSD, but not both. Therefore, when both PTSD and DSO symptoms were present, CPTSD was coded as present for this participant. The ITQ has been validated in numerous languages and is the most commonly used measure for assessing CPTSD [22,23,24,25]. In our sample, excellent internal consistency was observed for the ITQ total score (α = 0.90) and good internal consistencies for the PTSD (α = 0.89) and DSO (α = 0.87) subscales.

#### 2.2.3. Psychological Distress, Depression, and Anxiety (BSI-18, PHQ-9, GAD-7)

Psychological distress was evaluated using the Brief Symptom Inventory (BSI-18) [26], which comprises 18 items rated on a four-point Likert scale, ranging from ‘not at all’ to ‘very often’. This assessment yields a total score as well as three subscale scores (depression, anxiety, somatization). Previous studies have reported good reliability and validity for both the subscales and the total score [26,27]. In our sample, good to excellent internal consistency was observed (total score α = 0.93; depression = 0.87, somatization = 0.84, anxiety = 0.85). The PHQ-9 [28] is the depression module of the Health Questionnaire, which uses nine questions to assess the presence and severity of depressive symptoms. The symptoms assessed are based on the nine depression criteria of the DSM-IV, which can both support the diagnosis of depression and evaluate the severity of depressive symptoms. The PHQ-9 is the most commonly used measure to screen for depression, and good clinical utility has been reported for it [29,30,31]. In our sample, an excellent internal consistency was observed for the PHQ-9 total score (α = 0.91). The Generalized Anxiety Disorder Scales GAD-7 is a self-report questionnaire for the assessment of anxiety [32]. It consists of seven items to assess generalized anxiety disorder in line with symptoms described in the DSM-IV. Good psychometric properties and clinical usability have been reported for the German GAD-7 [33,34]. In our sample, an excellent internal consistency was observed for the GAD-7 total score (α = 0.91).

#### 2.2.4. Work Ability Index (WAI)

Participants were requested to assess their perceived working ability using a single item drawn from the Work Ability Index (WAI) [35]. This item allowed ratings on a scale from 0 (‘very poor working ability’) to 4 (‘very good working ability’). The WAI is widely employed across various research domains as a tool for evaluating working ability [35].

#### 2.2.5. Operationalized Psychodynamic Diagnosis Structure Questionnaire—Short Form (OPD-SQS)

The Operationalized Psychodynamic Diagnosis Structure Questionnaire—Short Form (OPD-SQS) [36] is a 12-item self-assessment tool designed to gauge the level of personality functioning. It generates a total score ranging from 0 to 48, along with scores for each of its three subscales (self-perception, interpersonal contact, and relationship model), ranging from 0 to 16. Higher scores indicate more pronounced impairments in personality functioning. Previous research has documented good validity and reliability for the OPD-SQS [37]. Within our sample, we observed good to excellent internal consistency for both the total scale (α = 0.91) and the three subscales (α = 0.81–0.87).

#### 2.2.6. Epistemic Trust, Mistrust, and Credulity Questionnaire (ETMCQ)

The Epistemic Trust, Mistrust, and Credulity Questionnaire (ETMCQ) is utilized to evaluate an individual’s capacity for epistemic trust [13]. It comprises 12 items assessing three subscales: ‘epistemic trust’, ‘mistrust’, and ‘credulity’, measured on a 7-point Likert scale. Responses range from ‘strongly disagree’ to ‘strongly agree,’ yielding a total score between 15 and 84. High levels of trust indicate an openness to social learning opportunities, while elevated mistrust signifies a propensity to view information sources as unreliable and resist external influence. Conversely, high credulity reflects uncertainty about one’s own stance, potentially heightening vulnerability to misinformation and exploitation by others [13]. The German version of the questionnaire has undergone validation in a representative sample, confirming its three-factor structure [38]. In our sample, a good internal consistency was found for the total scale (α = 0.81), as well as the trust (α = 0.81) and credulity subscale (α = 0.80), while the internal consistency of the mistrust (α = 0.69) subscale was questionable.

#### 2.2.7. Mental Health Treatment

To assess the mental health treatment history, participants were asked if they (a) had ever received any form of mental health treatment and, if yes, (b) if this treatment included psychotherapy. If participants had a history of mental health treatment, they were asked to specify if the treatment was an outpatient treatment, psychosomatic inpatient treatment, psychiatric inpatient treatment, or an outpatient clinic. Participants had the option for multiple answers. Finally, participants were asked to rate how helpful the treatment was for their mental health issue on a scale from ‘it got way better’ (1) to ‘it got way worse’ (5). The answers were summarized as ‘improvement’ (scores 1–2), ‘no improvement (score 3)’, and ‘deterioration’ (scores 4–5).

### 2.3. Statistics

Data were included for participants with complete ITQ datasets. Sociodemographic data are presented for both groups: those who met the CPTSD criteria according to the ITQ and those who did not. To account for the low number of gender-diverse individuals, the variable sex was dichotomized into 1 = male gender and 2 = non-male gender for all analyses except descriptive analyses. Prevalence rates of the ITQ symptom cluster are presented for the total sample as well as the subsamples of participants with and without a history of mental health treatment. Mental health treatment uptake and treatment satisfaction were compared between three groups: (a) participants who fulfilled the CPTSD criteria according to the ITQ, (b) participants who fulfilled the PTSD criteria according to the ITQ, and (c) participants who fulfilled the criteria for a depressive disorder according to the PHQ-9 and BSI-18. If participants fulfilled both PTSD and CPTSD criteria, they were assigned to the CPTSD group. To be assigned to the depression group, participants had to score above the recommended clinical cut-off for both PHQ-9 and BSI-18. If participants fulfilled both the depression and CPTSD or PTSD category, they were assigned to the CPTSD or PTSD group, accordingly. Finally, the level of psychological distress and personality functioning was also compared between the three groups with analyses of variance (ANOVA). To account for potential confounders, analyses were corrected for all sociodemographic variables with statistically significant differences in the univariate analyses. Thus, age, working status (dummy variable: employment vs. else), and family income (dummy variable: below EUR 1.250 vs. above EUR 1.250) were added as covariates. Effect sizes of group differences were estimated by calculation of partial eta square (*η*^2^), with values of *η*^2^ = 0.01 considered small, *η*^2^ = 0.06 medium, and *η*^2^ = 0.14 large effect sizes, respectively [39]. *p*-values < 0.05 were considered statistically significant, and all calculations were conducted using IBM SPSS (v21.0).

## 3. Results

### 3.1. Sociodemographic Data

A total of n = 2519 participants were initially included in the study. Due to missing data on the ITQ, n = 601 (23.9%) participants were excluded, and the remaining n = 1918 individuals were included in the analyses. While excluded participants were significantly younger (46.6 vs. 51.5 years; *p* < 0.001; *η*^2^ = 0.014) and reported significantly less ACEs (0.6 vs. 1.0; *p* < 0.001; *η*^2^ = 0.009) than included participants, the differences were of negligible to small effect size. No significant differences were found in terms of gender (*p* = 0.95).

Of the included participants, n = 29 (1.5%) fulfilled the diagnostic criteria for CPTSD, n = 51 for PTSD (2.7%), and n = 79 (4.1%) for depressive disorders. Overall, participants with CPTSD were significantly younger and more frequently unemployed than individuals without CPTSD, while participants with depression reported the lowest mean monthly family net income and were most often retired. No significant differences were observed in terms of gender, marital status, migration background, or the highest level of education. For details, see Table 1.

### 3.2. Clinical Data

In the total sample of the general population, n = 29 (1.5%) of the participants fulfilled the criteria for CPTSD and n = 51 (2.7%) for PTSD. However, the prevalence for the individual PTSD (i.e., re-experiencing, avoidance, and persistent perception of heightened current threat) and DSO (affective dysregulation, negative self-concept, and problematic relationships) core symptoms were particularly higher, with the highest rates for affective dysregulation (30.1%). Participants with a depressive disorder showed high rates of present core symptoms, specifically for DSO key symptoms: 82.3% of participants with depressive disorders reported problematic relationships, 73.4% affective dysregulation, and 60.8% a negative self-concept. For details, see Figure 1.

In the total sample, a better epistemic stance was significantly associated with lower levels for all PTSD and DSO symptom clusters (*p* < 0.001). However, correlations between the epistemic stance and DSO symptom cluster were particularly higher (r = −0.33–r = −0.35) than with the PTSD symptom cluster (r = −0.10–r = −0.23). Epistemic trust was not significantly associated with symptoms of re-experiencing and avoidance but showed a negative association with persistent perception of heightened current threat, affective dysregulation, negative self-concepts, and problematic relationships. While epistemic mistrust was associated with all PTSD and DSO symptoms, particularly lower correlations were found with intrusive symptoms and avoidance behavior. The highest correlations with all PTSD and DSO subscales were found for epistemic credulity. For details, see Table 2.

### 3.3. Trauma Characteristics of Participants with CPTSD

The most frequently reported form of trauma among the n = 29 individuals with CPTSD was sexual abuse (n = 6, 20.7%), followed by severe injury, illness, or death of a loved one (n = 5, 17.2%); suffering from a psychiatric disorder (n = 5, 17.2%) or a physical disease (n = 4, 13.8%); loss of job or severe financial distress (n = 3, 10.3%); experiencing a natural disaster or a terror attack (n = 2, 6.9%); personally experiencing injury, robbery, or theft (n = 2, 6.9%); and experiencing divorce or severe relationship problems (n = 2, 6.9%). About one-third of the participants had experienced trauma within the last year (n = 11, 37.9%), while 24.1% (n = 7) reported that the incidence had happened 1–5 years ago, 10.3% (n = 3), 5–10 years ago, and 24.1% (n = 7) reported that the incidence happened over 10 years ago.

### 3.4. Mental Health Treatment Uptake

Of all the included participants, a total of n = 252 (13.1%) reported to have received mental health treatments in the past. About two-thirds (65.5%) of participants who fulfilled the diagnostic criteria for CPTSD had received mental health treatment. In comparison, a significantly lower proportion of participants with PTSD (41.2%) or with depression (31.6%) had received mental health treatment (χ^2^ = 10.654, *p* = 0.031). Among participants who fulfilled the CPTSD criteria, 90% had received psychotherapy. The most frequent form of treatment was outpatient psychotherapy (88.2%), followed by psychiatric inpatient treatments (47.4%), psychotherapy outpatient clinic treatment (44.4%), and psychosomatic inpatient treatment (36.8%). Compared to participants with PTSD and depression, individuals with CPTSD had been in psychiatric inpatient treatment and psychotherapy outpatient clinics significantly more often, while no significant difference was observed for outpatient psychotherapy and psychosomatic inpatient treatments. For details, see Table 3 and Figure 2.

Overall, 83.6% of participants who had received mental health treatment felt that their symptoms had improved due to the treatment, while 15.0% had experienced no change in their symptoms and 1.4% even symptom deterioration. However, when the sample was split into the three groups, it became apparent that participants who fulfilled the CPTSD criteria reported worse treatment satisfaction (χ^2^ = 17.301, *p* = 0.008): 52.9% reported symptom improvement, while 41.2% experienced no change and 5.9% symptom deterioration. For details, see Figure 3.

### 3.5. Comorbidity and Personality Functioning

The sample was further divided into three groups to compare the level of psychological distress (BSI-18, PHQ-9, GAD-7) and personality functioning (OPD-SQS, ETMCQ): group (A) consisted of n = 29 participants who fulfill the CPTSD criteria; group (B) included participants fulfilled the PTSD criteria; and (C) participants who fulfilled criteria for depressive disorders. Participants with CPTSD reported a significantly higher number of ACEs than participants with PTSD, as well as significantly higher scores for depression, anxiety, and worse personality functioning (as measured by the OPD-SQS) and worse epistemic mistrust and credulity than participants with PTSD. Additionally, participants with CPTSD also reported significantly higher depression and anxiety scores than patients with depression. Regarding the subjective working ability, participants with depression reported significantly worse scores than participants with CPTSD and PTSD, while there was no significant difference between participants with CPTSD and PTSD. For details, see Table 4.

## 4. Discussion

The primary objective of this study was to provide insights into the utilization of mental health treatment and the satisfaction of individuals with CPTSD in a large, representative sample from the German general population. As a second aim, we investigated comorbidity rates for depression, anxiety, and various aspects of personality functioning in the entire population-based sample.

We identified a prevalence of 1.5% for CPTSD in contrast to 2.7% for PTSD, which is slightly higher than previously reported prevalence rates in the German general population [4]. This increase may be attributed to differences in the applied version of the ITQ questionnaire, as well as elevated psychosocial distress related to the COVID-19 pandemic [40]. This is reflected in the higher prevalence of natural stated disasters as the traumatic experience in our study (6.9%), compared to the above-mentioned German studies pre-COVID-19 (2018: 3.5%). Furthermore, we observed relatively high rates of individual PTSD and DSO core symptoms, particularly among participants with a history of mental health treatments. Although only 6.1% of all participants met the criteria for a self-reported PTSD or CPTSD diagnosis, 53.2% of those with a mental health treatment history reported scores exceeding the clinical cut-off for affective dysregulation and 43.3% for severely problematic relationships. Additionally, more than one-third of these individuals reported at least one core PTSD symptom (re-experiencing: 38%; avoidance: 36.5%; persistent perception of heightened current threat: 33.7%). These findings underscore the significant prevalence of trauma-related symptoms among individuals with mental health concerns, even when the full diagnostic criteria for PTSD are not met. This aligns with previous research indicating that trauma-related symptoms are often underestimated and underdiagnosed in patients with mental health issues [41,42]. Routinely using recommended and validated screening tools such as the ITQ [23] in mental health treatment might allow us to identify patients with trauma-related symptoms early on.

In our study, we found that higher symptoms of CPTSD were associated with poorer epistemic trust. Prior research has indicated that psychopathologies are linked to a compromised epistemic stance and, concurrently, disruptions in the social learning process it facilitates [10]. In contrast, restoring epistemic trust and reducing epistemic mistrust have been shown to improve trauma-related symptoms in patients with CPTSD [17]. Recent studies have demonstrated a direct correlation between epistemic disruption and treatment outcomes in psychosomatic rehabilitation [43,44]. Psychotherapy has been suggested as a method to disrupt the cycle of epistemic mistrust and credulity by nurturing epistemic trust [9,45,46]. Understanding the significance of epistemic trust and mistrust can assist clinicians in better comprehending potential disruptions during a patient’s treatment. This can be achieved by: (1) examining potential factors that might exacerbate mistrust, (2) directly addressing the unique nature of these epistemic dynamics with each patient (for instance, during treatment: “What could I do to help us better understand when you feel you cannot trust me?” or “Could you help me identify instances where I may have said or done something that could undermine your trust in me?”), (3) communicating to the patient that these difficulties are acknowledged, and (4) consequently, establishing a shared focus for both the patient and psychotherapist to revisit this understanding when a rupture occurs.

It can be assumed that the difficulties in forming and maintaining relationships, repeated breakdowns of relationships, social withdrawal, and persistent distrust of others—often observed in patients with severe interpersonal traumata [6]—may be linked to a lack of epistemic trust in social relationships. Consequently, there may be a markedly impaired capacity for social learning in benign social settings, such as psychotherapeutic treatment [11]. The ability to assess the trustworthiness, relevance, and generalizability of information from others forms the basis for building a trusting relationship between the patient and their therapist. This fosters an environment that encourages openness for knowledge exchange and the exploration of diverse perspectives and viewpoints [9]. Hence, recognizing and addressing challenges related to mentalizing and disruptions in epistemic processes in patients with traumatic experiences may be pivotal in facilitating change and recovery [11,47].

In our study, approximately two-thirds of participants with CPTSD had undergone mental health treatment, with almost all of them receiving psychotherapy. This finding is quite encouraging and is likely attributed to the structure of the German healthcare system, where mental health treatment, including psychotherapy, is fully covered by healthcare providers. However, this is not the case in many other European countries. For instance, in neighboring Austria, psychotherapy is typically not covered by general health insurance [48]. An analysis of over 6000 patients from a psychosomatic rehabilitation center in Austria revealed that less than half of the patients received financial support from their health insurance and were unlikely to afford psychotherapy out of pocket [49]. Given the relatively high prevalence of patients with PTSD or CPTSD among psychosomatic inpatients [50], it can be presumed that psychotherapeutic care for these patients is significantly less accessible in several European countries.

Overall, the vast majority (more than 80%) of participants with a history of mental health treatment reported experiencing benefits from therapy. Only 15% reported no change, and 1.4% reported symptom deterioration. These rates are consistent with previously reported success rates for psychotherapy treatments [51,52]. However, when comparing the treatment satisfaction of participants with CPTSD to participants with PTSD or depression, a substantial and statistically significant difference emerged. Participants with a depressive disorder or PTSD reported symptom improvement in approximately 80%, while 15–22% reported no change, and 0–5% reported a deterioration in symptoms. In contrast, for patients with CPTSD, only 53% reported symptom improvement, while 41% reported no change, and 6% reported a deterioration in symptoms. While there is only limited data comparing mental health outcomes for patients with and without CPTSD, a review of psychotherapeutic interventions for CPTSD has shown that traditional interventions were associated with poorer treatment outcomes than in other forms of trauma-related disorders [16]. The results of our study thus highlight the need for specific and tailored interventions for patients with CPTSD, such as Skills Training in Affect and Interpersonal Regulation Narrative Therapy (SNT) [53], trauma-focused psychodynamic therapy [54,55], or specific mentalization-based treatments [11].

### Strengths and Limitation

The study has several strengths and limitations. For one, to our knowledge, this is the first study to investigate the mental health treatment uptake of individuals with CPTSD in a representative sample from the general population. Secondly, this is the first study to link these data to the novel concept of epistemic trust, potentially enhancing our understanding of the lower treatment satisfaction observed in patients with severe disorders like CPTSD. Nevertheless, the cross-sectional nature of the data used in our study restricts our ability to explore therapy outcomes comprehensively, hindering an in-depth analysis of predictors for treatment satisfaction and outcome. Unfortunately, a substantial proportion of participants was excluded due to missing ITQ data, which limits the findings of this study. However, differences between included and excluded participants were of negligible to small effect sizes. Furthermore, the classification of CPTSD cases relies solely on self-ratings by patients, which was not corroborated by clinical ratings. While the ITQ is recommended as a screening instrument for CPTSD, it does not replace a clinician’s evaluation. Unfortunately, due to the nature of data collection, we lack detailed information on the type, length, and frequency of mental health treatment, thereby limiting the interpretation and discussion of our findings. Additionally, no information on psychopharmacological treatment was collected.

## 5. Conclusions

Our study’s data reveals that, although only a small proportion of the German general population exhibits clinical signs of CPTSD, affected individuals experience a significant symptom burden and substantial comorbidities. Despite the majority of participants with CPTSD having undergone mental health treatment, their satisfaction with the treatment was unsatisfactory. One potential explanation for this lies in heightened levels of epistemic mistrust and epistemic credulity, hindering the establishment of trusting relationships and impeding social learning. This, in turn, may compromise the effectiveness of psychotherapeutic interventions. Based on our study results, we recommend routine screening for CPTSD symptoms, regular assessment of epistemic trust, and the implementation of tailored interventions for individuals with CPTSD.

## Figures and Tables

**Figure 1 jcm-13-02735-f001:**
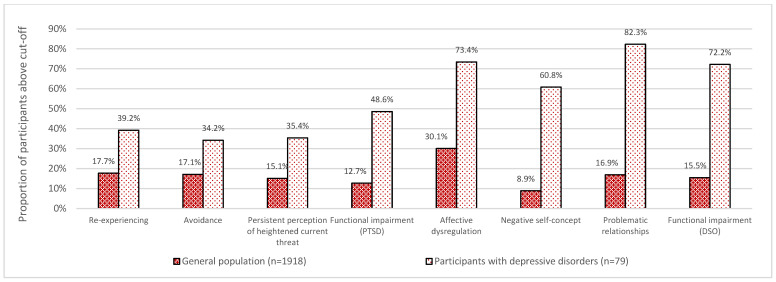
Complex post-traumatic stress disorder (CPTSD) symptoms in the general population and for participants with depressive disorders.

**Figure 2 jcm-13-02735-f002:**
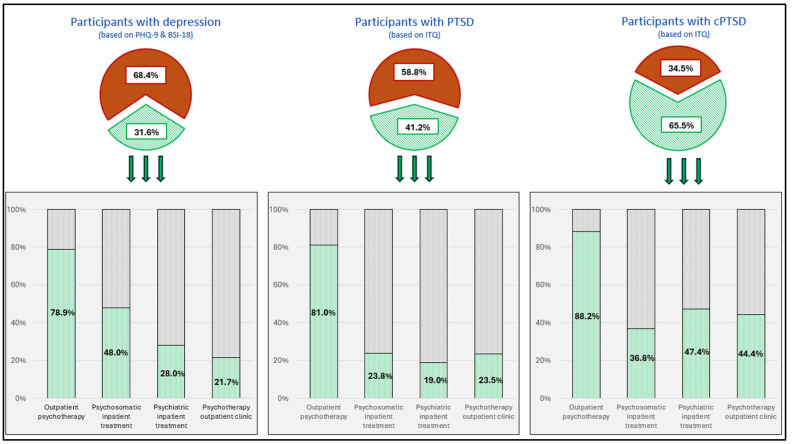
Proportion of mental health treatment uptake by participants with depression (based on the PHQ-9 and BSI-18 cut-offs), post-traumatic stress disorder (PTSD) (based on the ITQ cut-off), and complex post-traumatic stress disorder (CPTSD). The red proportion of the circle indicates no mental health treatment, while the green proportion shows the percentage of participants from this group who had participated in mental health treatment. The graphs below the circle show the proportion of participants from this group who participated in the different forms of mental health treatments (the sum of mental health treatment uptake for the different treatment forms may exceed 100% due to the possibility of multiple answers).

**Figure 3 jcm-13-02735-f003:**
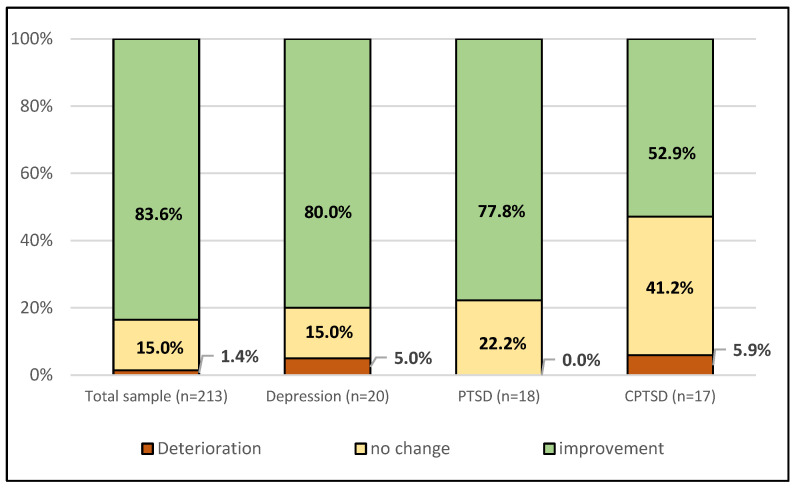
Proportion of participants with a history of mental health treatment who reported improvement, no change, or deterioration of their symptom severity, stratified for participants with depression, post-traumatic stress disorder (PTSD), and complex post-traumatic stress disorder (CPTSD).

**Table 1 jcm-13-02735-t001:** Sociodemographic data of the total sample as well as for participants with CPTSD, PTSD, and depression.

	Total Sample (n = 1918)		Participants with Depression (n = 79)		Participants with PTSD (n = 51)		Participants with CPTSD (n = 29)			
	n	%	n	%	n	%	n	%	χ^2^	*p*-Value
**Gender**									2.369	0.67
male	905	47.2%	33	41.8%	19	37.3%	11	37.9%		
female	1010	52.7%	46	58.2%	31	60.8%	18	62.1%		
diverse	3	0.2%	0	0.0%	1	2.0%	0	0.0%		
**Age**									28.805	0.001
<30 years	303	15.8%	9	11.4%	12	23.5%	5	17.2%		
30–39 years	239	12.5%	5	6.3%	9	17.6%	10	34.5%		
40–49 years	290	15.1%	10	12.7%	3	5.9%	4	13.8%		
50–59 years	405	21.1%	14	17.7%	9	17.6%	6	20.7%		
60–60 years	335	17.5%	14	17.7%	11	21.6%	3	10.3%		
>70 years	346	18.0%	27	34.2%	7	13.7%	1	3.4%		
**Marital Status**									7.929	0.44
married	880	45.9%	27	34.2%	23	45.1%	13	44.8%		
single	545	28.4%	24	30.4%	17	33.3%	11	37.9%		
divorced	275	14.3%	14	17.7%	5	9.8%	3	10.3%		
widowed	211	11.0%	14	17.7%	5	9.8%	2	6.9%		
missing	7	0.4%	0	0.0%	1	2.0%	0	0.0%		
**Living with partner**									6.677	0.15
living with partner	1101	57.4%	34	43.0%	27	52.9%	16	55.2%		
not living with partner	786	41.0%	45	57.0%	24	47.1%	12	41.4%		
Missing	31	1.6%	0	0.0%	0	0.0%	1	3.4%		
**Migration background**									1.837	0.40
No	1539	87.3%	67	84.8%	46	90.2%	23	79.3%		
Yes	220	12.7%	12	15.2%	5	9.8%	6	20.7%		
**Monthly family net household income**									17.001	0.030
EUR < 1500	405	21.1%	33	41.8%	10	19.6%	7	24.1%		
EUR 1500–2500	611	31.9%	28	35.4%	14	27.5%	10	34.5%		
EUR 2500–3500	437	22.8%	6	7.6%	14	27.5%	7	24.1%		
EUR > 3500	420	21.9%	10	12.7%	11	21.6%	5	17.2%		
missing	45	2.3%	2	2.5%	2	3.9%	0	0.0%		
**Highest level of education**									8.582	0.38
school not finished/still in school	77	4.0%	5	6.3%	3	5.9%	2	6.9%		
compulsory school	539	28.1%	33	41.8%	12	23.5%	10	34.5%		
higher education	1097	57.2%	35	44.3%	30	58.8%	17	58.6%		
university degree	189	9.9%	4	5.1%	5	9.8%	0	0.0%		
missing	16	0.8%	2	2.5%	1	2.0%	0	0.0%		
**Employment**									30.430	0.002
full-time employment	796	41.5%	14	17.7%	21	41.2%	11	37.9%		
part-time employment	238	12.4%	7	8.9%	9	17.6%	3	10.3%		
unemployed	155	8.1%	15	19.0%	3	5.9%	9	31.0%		
in training	102	5.3%	4	5.1%	4	7.8%	2	6.9%		
retired	604	31.5%	38	48.1%	13	25.5%	3	10.3%		
missing	23	1.2%	1	1.3%	1	2.0%	1	3.4%		

CPTSD = complex post-traumatic stress disorder; PTSD = post-traumatic stress disorder.

**Table 2 jcm-13-02735-t002:** Correlations of International Trauma Questionnaire (ITQ) and Epistemic Trust, Mistrust, and Credulity Questionnaire (ETMCQ) scales.

	ETMCQ Total	ETMCQ Trust	ETMCQ Mistrust	ETMCQ Credulity
Re-experiencing	−0.10 ***	0.01	0.07 **	0.17 ***
Avoidance	−0.13 ***	0.01	0.09 ***	0.19 ***
Persistent perception of heightened current threat	−0.23 ***	−0.09 ***	0.18 ***	0.24 ***
Functional impairment (PTSD)	−0.20 ***	−0.07 **	0.17 ***	0.21 ***
Affective dysregulation	−0.35 ***	−0.15 ***	0.30 ***	0.33 ***
Negative self-concept	−0.33 ***	−0.16 ***	0.24 ***	0.35 ***
Problematic relationships	−0.35 ***	−0.15 ***	0.28 ***	0.35 ***
Functional impairment (DSO)	−0.30 ***	−0.11 ***	0.25 ***	0.31 ***

DSO = Disturbances in self-organization; ** *p* < 0.01, *** *p* < 0.001.

**Table 3 jcm-13-02735-t003:** Mental health treatment uptake of participants with CPTSD, PTSD, and depression.

	**Total Sample** **(n = 1918)**		**CPTSD** **(n = 29)**		**PTSD** **(n = 51)**		**Depression** **(n = 79)**			
	**n**	**%**	**n**	**%**	**n**	**%**	**n**	**%**	**χ^2^**	** *p* ** **-Value**
Mental health treatment ^1^	252	13.1%	19	65.5%	21	41.2%	25	31.6%	138.943	<0.001
Psychotherapy ^2^	218	11.4%	17	58.6%	18	35.3%	21	26.6%	123.590	<0.001
Missing	52	2.7%	1	3.4%	1	2.0%	2	2.5%		
	**Total Sample** **(n = 252)**		**CPTSD** **(n = 19)**		**PTSD** **(n = 21)**		**Depression** **(n = 25)**		**χ^2^**	** *p* ** **-value**
If yes, which type of therapy?										
Outpatient psychotherapy	184	73.0%	15	88.2%	17	81.0%	15	78.9%	1.592	0.66
missing	39	15.5%	2	10.5%	3	14.3%	4	16.0%		
Psychosomatic inpatient treatment	56	22.2%	7	36.8%	5	23.8%	12	48.0%	3.949	0.27
missing	56	22.2%	2	10.5%	7	33.3%	5	20.0%		
Psychiatric inpatient treatment	51	20.2%	8	47.4%	4	19.0%	7	28.0%	8.908	0.031
missing	56	22.2%	2	10.5%	7	33.3%	5	20.0%		
Psychotherapy outpatient clinic	45	19.6%	8	44.4%	4	23.5%	5	21.7%	8.134	0.043
missing	35	15.2%	2	11.1%	3	17.6%	4	17.4%		

CPTSD = complex post-traumatic stress disorder; PTSD = post-traumatic stress disorder; ^1^ any kind of mental health treatment, including psychotherapy; ^2^ specifically receiving psychotherapy.

**Table 4 jcm-13-02735-t004:** Clinical data for participants with complex post-traumatic stress disorder (CPTSD), post-traumatic stress disorder (PTSD), and depression.

	Group A: CPTSD (n = 29)		Group B: PTSD (n = 51)		Group C: Depression (n = 79)			
	Mean	SD	Mean	SD	Mean	SD	*p*-Value	*η* ^2^
ACE	3.7	2.7	2.0	2.5	2.8	2.8	0.043 ^a^	0.04
BSI-18								
Depression	11.3	6.5	5.9	5.2	10.3	3.8	<0.001 ^a,c^	0.14
Anxiety	8.7	5.3	4.5	4.7	6.3	3.9	<0.001 ^a,b^	0.09
Somatization	6.5	5.6	4.2	4.0	6.1	4.5	0.031 ^a^	0.05
PHQ-9 total score	14.7	6.8	8.2	5.0	11.9	3.8	<0.001 ^a,b,c^	0.18
GAD-7 total score	11.8	5.2	7.6	4.7	8.7	3.5	<0.001 ^a,b^	0.10
WAI	5.2	2.7	5.6	2.5	4.0	2.5	0.43	0.01
OPD-SQS								
total score	41.6	8.0	32.3	9.4	36.5	8.8	<0.001 ^a,c^	0.11
self-perception	12.9	3.4	10.0	3.9	10.7	4.0	0.005 ^a^	0.07
interpersonal contact	13.6	3.0	9.8	3.4	12.2	3.2	<0.001 ^a,c^	0.14
relationship model	15.1	3.3	12.5	3.9	13.6	3.5	0.040 ^a^	0.04
ETMCQ								
Trust	21.5	5.9	23.9	4.8	22.2	6.0	0.16	0.02
Mistrust	23.8	5.1	20.4	5.2	20.8	5.7	0.043 ^a^	0.04
Credulity	22.5	4.8	18.8	4.8	19.3	5.6	0.018 ^a^	0.05

ACE = adverse childhood experiences; BSI-18 = Brief Symptom Inventory; PHQ-9 = Patient Health Questionnaire depression module; GAD-7 = General Anxiety Disorder Questionnaire; WAI = Working Ability Index; OPD-SQS = Operationalized Psychodynamic Diagnosis Structure Questionnaire—Short Form; ETMCQ = Epistemic Trust, Mistrust, and Credulity Questionnaire. Higher scores on the WAI and ETMCQ trust scale indicate better functioning, while higher scores on all other scales indicate higher symptom severity and worse personality functioning. All analyses were corrected for age, working status (dummy variable: employment vs. else), and family income (dummy variable: below EUR 1.250 vs. EUR above 1.250). ^a^ Bonferroni corrected *p* < 0.05 between CPTSD and PTSD; ^b^ Bonferroni corrected *p* < 0.05 between CPTSD and depression; ^c^ Bonferroni corrected *p* < 0.05 between PTSD and depression. Effect size values *η*^2^ ≥ 0.01 were considered small, *η*^2^ ≥ 0.06 as medium, and *η*^2^ ≥ 0.14 as large.

## Data Availability

The study data are available from the corresponding author upon reasonable request.

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
