# Peer review of "Utilization of Mental Health Provision, Epistemic Stance and Comorbid Psychopathology of Individuals with Complex Post-Traumatic Stress Disorders (CPTSD)—Results from a Representative German Observational Study"

_jcm, 2024, doi:10.3390/jcm13102735_

Round 1
Reviewer 1 Report
Comments and Suggestions for Authors
Review: Utilization of mental health provision, epistemic stance and comorbid psychopathology of individuals with complex post-traumatic stress disorders (CPTSD) – results from a representative German observational study
This paper evaluates information on mental health provision uptake, satisfaction with treatment, personality functioning, and comorbidities of individuals with CPTSD in a representative sample of the German general population using a range of relevant questionnaires.
It is relevant for several fields, such as psychology, psychiatry, and public health.
The study is well written. However, there are some parts that could be improved upon.
1.
The authors write: “Complex post-traumatic stress disorder (CPTSD) is a severely debilitating symptom cluster which has been added to the International Classification of Diseases (ICD-11). The aim of this study was to evaluate information on mental health provision uptake, treatment satisfaction, personality functioning and comorbidities of individuals with CPTSD in a representative sample of the German general population.” (lines 28 to 32)
While the first statement implies the importance of researching CPTSD, the next sentence is somewhat disconnected. It would be beneficial for the reader to add another short sentence after the first one regarding something about the earlier research on the topic or the research gap that this article will fill (for example, in the abstract it could be included that, as authors wrote later on, “information regarding psychotherapy uptake and treatment satisfaction is still scarce”).
2.
The authors write: “A traumatic event is defined as an extremely threatening or horrific event or series of events that can result in symptoms of Post-Traumatic Stress Disorder (PTSD) and an additional symptom complex in self-organization.” (lines 52 to 55)
I would clarify what exactly was meant with “symptom complex in self-organization”. “Self-organization” is a term that can be interpreted in various ways.
3.
The authors write: “… which is associated with ineffective mentalizing alongside social trauma – may cause a state of epistemic hypervigilance …” (lines 72 to 73)
It would be beneficial to provide a brief explanation of the terms “mentalizing” and “epistemic hypervigilance” since broader audience, not necessarily specialized in the relevant fields, might not be familiar with it.
4.
The authors write: “By administering face-to-face interviews as well as self-report questionnaires to randomly selected persons within predefined regions, a total of N=2,519 participants could be included.” (lines 105 to 107)
To the reader it is not completely clear from this sentence whether all the 2,519 participants were interviewed and filled out the questionnaires or whether some were interviewed and some filled out the questionnaires.
5.
The authors write: “Inclusion criteria comprised sufficient German language skills, an age > 16 and informed consent (for minors, informed consent was additionally obtained from a parent/legal guardian).” (lines 107 to 110)
I suggest writing “being older than 16”.
6.
The authors wrote: “Symptom were coded as present if a score of 2 or higher was given by the participant. A diagnosis of PTSD was given if at least one of two symptoms of each of the three PTSD clusters were coded as present plus at least one functional impairment cluster associated with PTSD was given. A diagnosis of CPTSD was given if the diagnostic criteria for PTSD were fulfilled and at least one of two symptoms of each of the three DSO clusters were coded as present plus at least one functional impairment cluster associated with CPTSD was given [15].” (lines 149 to 155)
I would rephrase this text for clarity. The use of the phrase “was given”, referencing the authors at times and the participants at other times, is confusing to the reader.
7.
The authors wrote: “Finally, participants were asked to rate how helpful the treatment was for their mental health issue on a scale from ‘it got way better’ (1) to ‘it got way worse’ (5). The answers were summarized as ‘improvement’, ‘no improvement’ and ‘deterioration’.” (lines 210 to 213)
Even though the sentence is in general quite clear, a reader might benefit from further clarification regarding how exactly the mentioned categories (improvement, no improvement, deterioration) relate to the scale.
8.
The authors wrote: “Regarding the subjective working ability, participants with depression reported significantly worse scores than participants with CPTSD and PTSD, while there was no significant difference between the two groups.” (lines 333 to 336)
It is not completely clear to the reader what the "two groups" refers to. Specifying clearly which two groups are being referred to here, would be beneficial.
9.
The authors wrote: “Sociodemographic data is presented for participants who did and did not fulfill the CPTSD criteria according to the ITQ.” (lines 216 to 218)
The sentence is somewhat unclear to the reader because of the formulation “did and did not”. I suggest rewriting it in a clearer way (for example: “for both groups: those who met the CPTSD criteria and those who did not”).
Also, “fulfil” should be “fulfil”.
10.
The authors wrote: “However, the prevalence for the individual core symptoms were particularly higher, with highest rates for affective dysregulation (30.1%).” (lines 254 to 256)
The reader would benefit from a clarification of what the “individual core symptoms” are. A short reference to them, as presented in the graph below, would suffice (written at the first mention of the “individual core symptoms” instead of at the end of the paragraph).
11.
The reader might benefit from the citation of the various questionnaires, mentioned in the article. Either that or they could be added in the supplementary materials. It would make it easier for the reader to find the exact items used in the paper.
12.
I suggest proofreading the paper as there are several inconsistencies typos in it. For example, the authors used two versions of the word (“fulfil” and “fulfil”). Sticking with one version is preferred.
Considering the above written, I suggest revising and resubmitting.
Comments on the Quality of English Language
Please, see above.
Reviewer 2 Report
Comments and Suggestions for Authors
This study investigated the mental health service use, epistemic stance, and comorbid psychopathology among individuals with probable CPTSD, PTSD, and depressive disorder in a representative study in Germany. This is a fascinating research topic. The authors have written the manuscript well, but the analytical plan seems less well-presented. The definition of mental health service use should be clearer. Please address my comments below and then I should be able to comment on the discussion section better. Thank you.
Abstract
1. Please provide the full name of the scale used upon its first appearance.
2. Use a consistent term for "mental health treatment uptake"; the authors sometimes use "mental health provision uptake," "Mental health uptake," and "therapy uptake."
Methods
1. Can the authors provide more information about how the sample was obtained? What are the "predefined regions"? How were these regions determined? What criteria were used?
2. The authors mentioned the psychometrics of the scale used in the past, but what is their performance in the current study?
3. Please revise the description of OPD-SQS. The author mentioned that there are three subscales, each on a 0-12 scale, but the total scores range from 0-48. Why not 0-36? This is unclear.
4. Similarly, for the ETMCQ, the author mentioned that there are 12 items, but the total scores range from 15-105. It is unclear how this was calculated. 5. Does mental health treatment also include pharmacological treatments? If so, please clarify.
6. Lines 222 and 223: (a) and (b) read the same; I believe the author means PTSD for (b).
7. It is suggested that the authors avoid using the term "diagnostic group," given that only self-reported rating scales were used to differentiate participants.
8. Were there any confounders for the observed differences between the three groups? If so, were they controlled with the ANCOVA models?
9. Is the analysis weighted for the population distribution? If not, does the current sample exhibit a similar composition to the population?
Results
1. The authors mentioned using 1 = male and 2 = non-male gender, but Table 1 shows three categories for sex. Please make these more consistent.
2. In Table 3, is it possible that participants received both mental health treatment and psychotherapy? How did the authors manage this?
Comments on the Quality of English LanguageThere are some uses of inconsistent terms otherwise the English writing is alright.
Reviewer 3 Report
Comments and Suggestions for Authors
Thank you for allowing me to read your work. I suggest the following points to make it much clearer and transparent:
1. In the introduction, expand on the relevance of the study, why it is important to address this issue, in this way, and why now.
2. In the methodology, it is important to report, as you did with the Adverse Childhood Experiences Questionnaire, the internal consistency of all the instruments used for the study sample.
3. The outcome "Mental health treatment" is not clear how it was validated to control biases among participants regarding the responses obtained and desirable within the research framework.
Round 2
Reviewer 2 Report
Comments and Suggestions for Authors
Thanks for the authors' revision. Wonder we can compute the internal consistency values for the ITQ and WAI too? Also, although the platform is designed for the collection of a representative sample of the German population, I wonder whether the data really demonstrate its representativeness?
